# Who is most at risk of dying if infected with SARS-CoV-2? A mortality risk factor analysis using machine learning of patients with COVID-19 over time: a large population-based cohort study in Mexico

Lauren D Liao [ID] ,[1] Alan E Hubbard,[1] Juan Pablo Gutierrez [ID] ,[2] Arturo Juárez-Flores,[2] Kendall Kikkawa,[3] Ronit Gupta,[4] Yana Yarmolich,[5] Iván de Jesús Ascencio-Montiel,[6] Stefano M Bertozzi [ID] [7,8,9]

For numbered affiliations see end of article.

**Correspondence to**
Lauren D Liao;
ldliao@berkeley.edu

## ABSTRACT

**Objective** COVID-19 would kill fewer people if health programmes can predict who is at higher risk of mortality because resources can be targeted to protect those people from infection. We predict mortality in a very large population in Mexico with machine learning using demographic variables and pre-existing conditions.

**Design** Cohort study.

**Setting** March 2020 to November 2021 in Mexico, nationally represented.

**Participants** 1.4 million laboratory-confirmed patients with COVID-19 in Mexico at or over 20 years of age.

**Primary and secondary outcome measures** Analysis is performed on data from March 2020 to November 2021 and over three phases: (1) from March to October in 2020, (2) from November 2020 to March 2021 and (3) from April to November 2021. We predict mortality using an ensemble machine learning method, super learner, and independently estimate the adjusted mortality relative risk of each pre-existing condition using targeted maximum likelihood estimation.

**Results** Super learner fit has a high predictive performance (C-statistic: 0.907), where age is the most predictive factor for mortality. After adjusting for demographic factors, renal disease, hypertension, diabetes and obesity are the most impactful pre-existing conditions. Phase analysis shows that the adjusted mortality risk decreased over time while relative risk increased for each pre-existing condition.

**Conclusions** While age is the most important predictor of mortality, younger individuals with hypertension, diabetes and obesity are at comparable mortality risk as individuals who are 20 years older without any of the three conditions. Our model can be continuously updated to identify individuals who should most be protected against infection as the pandemic evolves.

## STRENGTHS AND LIMITATIONS OF THIS STUDY

⇒ Compared with previous studies showing age and the presence of pre-existing conditions as important predictors of mortality, our study leveraged more powerful statistical approaches by examining mortality risk in a very large population.

⇒ Analyses based on a combination of machine learning and causal inference methods for prediction and estimation of risk factor impacts that is robust to model misspecification.

⇒ The phase analyses presented in this study demonstrate marked changes over time in the degree to which different risk factors predict mortality.

⇒ Pre-existing conditions are self-reported, and thus subject to potential misspecification.

⇒ Those who are included in the data may not be representative of the socioeconomic status of the entire population; thus, extrapolation of the result should be done with caution.

## INTRODUCTION

The probability of mortality associated with SARS-CoV-2 infection has varied enormously over time, among countries and among population groups within countries.[1] Interest in understanding who is at a higher risk of death has grown as this heterogeneity became more apparent. Identifying people at higher risk of severe disease and death will help health systems better respond and focus prevention resources on protecting them. We examine Mexico, a country with a very high-reported case fatality rate (4.7%) among those who have laboratory-confirmed COVID-19 as of 23 September 2022.[2]

Previous analyses in Mexico have found diabetes, obesity, hypertension,

immunosuppression and renal disease to be significant risk factors along with age and sex. Multiple authors have identified obesity and diabetes as important risk factors for mortality.[3–6] Escobedo-de la Peña *et al* also found a strong association with hypertension, which is consistent with results from Giannouchos *et al*.[6 7] Late-stage chronic kidney disease, although less prevalent, has also consistently been identified as a COVID-19 mortality risk factor. Older/male patients tend to have higher mortality risks than younger/female patients.[4 6 7] In a previous analysis, we found interactions between those comorbidities, suggesting a synergic effect when having more than one of diabetes, hypertension and obesity (larger OR when reporting the three conditions vs one or two).[8] We also found that the OR increased by age group with those over age 80 having 30-fold the risk of those aged 20–29.[8] One important consideration is that the prevalence of diabetes and hypertension is positively associated with age, so it has not been clear how this interaction is related to mortality risk. A more adaptive analysis performed by Martínez-Martínez *et al* developed a prediction model for severity of COVID-19, defined by hospitalisation and/or mortality. They examined the relationship of 14 variables with hospitalisation and mortality using interaction terms and splines to account for non-linear relationships.[9]

The pattern of age, sex and comorbidities being associated with higher mortality risk is not specific to Mexico, and the global literature on such associations is extensive. Researchers have identified old age, diabetes, obesity, chronic renal failure and congestive heart failure to be strongly associated with severe infection among both sexes in the Spanish population.[10] Researchers in Brazil showed that older age, male, kidney disease, obesity and/ or diabetes are strong predictors of mortality among other comorbidities such as chronic liver disease, immunosuppression and cardiovascular disease.[11 12] Another study used UK Biobank data and showed that pre-existing dementia, diabetes, chronic obstructive pulmonary disease (COPD), pneumonia and depression were positively associated with risk of hospitalisation and death.[13] An analysis from France found age, diabetes, hypertension, obesity, cancer, and kidney and lung transplants to be associated with risk of COVID-19-related hospitalisation and mortality, among others.[14] A Canadian study reported dementia, chronic kidney disease, cardiovascular disease, diabetes, COPD, severe mental illness, organ transplant, hypertension and cancer to be significant predictors of mortality.[15] Studies presented here are a non-exhaustive list of research studying COVID-19 risk factors and mortality. Recent meta-analyses and systematic reviews find significant mortality attributed to these pre-existing conditions.[1 16–18] Our goal in this study is to predict mortality using demographic factors and comorbidities, and to show how those predictions change over time in this rapidly evolving pandemic.

Although mortality risk estimation and risk factor identification have been examined in prior studies, we are concerned about the statistical validity and interpretation of the standard methods. A commonly used prediction tool, logistic regression, assumes a linear relationship of predictors against the log odds of mortality risk, but this logit-linear assumption will lead inevitably to biased estimates of risk (either underpredict or overpredict the risk) for subsets of the population. We instead used flexible, data-adaptive methods that can capture non-linearities in the dose–response, such as potential non-linear interactions between the predictors (eg, the potential interaction of age and diabetes on predicting death).[19 20] The better the model fits the study population, the more likely estimates are closer to the true joint relationship of mortality and risk factors.

We included pre-existing conditions, demographic variables, the Mexican state where the patient was treated and the month that the patient initiated care to fit our prediction algorithm. We conducted the analysis using an ensemble machine learning algorithm, super learner (SL), to form optimal combination of predictions from multiple machine learning methods.[19 20] We also estimated the comparative importance of variables for mortality risk prediction (holding all other variables constant) by non-parametrically estimating quantities inspired by causal parameters (parameters that compare so-called counterfactual distributions, in our case, causal relative risks (RR)). The statistical goal is to estimate and provide robust inference for impact estimates of the predictors without the arbitrary modelling assumptions that characterise the great majority of prior work.[21]

## METHODS
### Study population and design
The study population is drawn from the Mexican Social Security Institute (IMSS), a vertically integrated insurance and health system that provides coverage for over 60 million private sector employees and their families, including their parents, children and spouse. IMSS also provided care as part of the COVID-19 response for some non-beneficiaries, who are also included in the data set.

The data were recorded from 1 March 2020 to 3 November 2021 in a platform called SINOLAVE. They reflect the entire population of 4 482 292 patients who were registered as receiving care for suspected COVID-19 at an IMSS facility. The data set and the data entry process have been described previously.[22] The demographic variables include age, sex, insured by IMSS and indigenous status. The data contain pre-existing conditions reported by the patient or the family at presentation: asthma, cardiovascular disease, chronic liver disease, COPD, diabetes, haemolytic anaemia, HIV, hypertension, immunosuppression, neurological disease, obesity, cancer, renal disease and tuberculosis, as well as whether the patient currently smokes. Patients were asked at presentation about their pre-existing health conditions; these were not ascertained with reference to the patient's medical record, even for those patients insured by the IMSS. The data also include the Mexican state in which the patient

**Table 1** Summary table of baseline variables and pre-existing conditions

| | All time (March 2020 to November 2021) | Phase 1 (March 2020 to October 2020) | Phase 2 (November 2020 to March 2021) | Phase 3 (April 2021 to November 2021) |
|---|---|---|---|---|
| Sample size | 1 423 720 | 303 278 | 425 698 | 694 744 |
| Demographic variables | | | | |
| Age in years, mean (SD) | 42.15 (15.70) | 46.41 (16.04) | 44.89 (16.27) | 38.61 (14.34) |
| Sex=male (%) | 729 782 (51.3) | 158 248 (52.2) | 218 165 (51.2) | 353 369 (50.9) |
| Insured by IMSS=yes (%) | 1 358 440 (95.4) | 288 588 (95.2) | 402 754 (94.6) | 667 098 (96.0) |
| Indigenous=yes (%) | 7381 (0.5) | 2200 (0.7) | 1628 (0.4) | 3553 (0.5) |
| Pre-existing conditions | | | | |
| Hypertension=yes (%) | 228 901 (16.1) | 72 615 (23.9) | 83 735 (19.7) | 72 551 (10.4) |
| Diabetes=yes (%) | 169 869 (11.9) | 55 551 (18.3) | 61 120 (14.4) | 53 198 (7.7) |
| Obesity=yes (%) | 181 736 (12.8) | 55 965 (18.5) | 60 217 (14.1) | 65 554 (9.4) |
| Smoking=yes (%) | 87 161 (6.1) | 21 253 (7.0) | 28 346 (6.7) | 37 562 (5.4) |
| Asthma=yes (%) | 25 297 (1.8) | 7951 (2.6) | 7765 (1.8) | 9581 (1.4) |
| Renal disease diagnosis=yes (%) | 24 099 (1.7) | 8912 (2.9) | 8555 (2.0) | 6632 (1.0) |
| Outcome | | | | |
| Death=yes (%) | 149 805 (10.5) | 53 530 (17.7) | 62 517 (14.7) | 33 758 (4.9) |

IMSS, Mexican Social Security Institute.

received care, COVID-19 test results (from both PCR tests and antigen tests), the month that the patient initiated care and COVID-related mortality. The outcome, death, is ascertained as COVID-related mortality within this study period between March 2020 and November 2021; we only consider deaths after patients initiated care. In addition, we extracted a different data set from the National Council of Science and Technology to determine the dominant circulating variant in each month.[23] A short summary can be found in table 1 (online supplemental table S1). We define COVID-19 positive as a positive PCR or antigen test.

From the full data set, we generated an analytical sample (n=1 423 720) (online supplemental figure S1). We exclude those under the age of 20 years, those without any positive COVID-19 test result from either the PCR or antigen tests and those with unknown pre-existing conditions. We also created a variable that corresponds to the phase changes in the epidemic curve: phase 1 is from 1 March 2020 to 31 October 2020; phase 2 is from 1 November 2020 to 31 March 2021; and phase 3 is from 1 April 2021 to 3 November 2021 as previously described.[22]

### Patient and public involvement

Patients or the public were not involved in the design, or conduct, or reporting, or dissemination plans of our research.

### Statistical analysis

#### Mortality risk prediction using SL

We predict mortality risks with SL[19 20] using predictors: pre-existing conditions, demographic variables, the Mexican state where the patient was treated and the month that the patient initiated care. SL combines a set of user-supplied machine learning algorithms, which includes both simple, parametric fits and flexible algorithms, and three-fold cross validation was used to create an optimally weighted combination. This optimal fit is found by creating a combination of algorithms that minimise the cross-validated risk (in our case, the negative log-likelihood). SL has the property that asymptotically it will perform at least as well as the best fitting algorithm in the library.[19 20] Thus, it is important to include a diverse and large set of learners as candidates to ensure the model can fit complex patterns if warranted, but also simpler parametric models if simpler fits are sufficient. The following learners were included in the SL library: Bayesian additive regression trees,[24] Bayesian generalised linear model,[25] elastic net regression,[26] empirical mean, generalised additive model,[27] least absolute shrinkage and selection operator regression,[28] logistic regression, multivariate adaptive regression splines,[29] random forest,[30] ridge regression[31] and extreme gradient boosting algorithms.[32] We estimate the prediction performance, via the area under the receiver operating characteristic curve (AUC), and derive a 95% CI for the estimated AUC.[33] We compare the SL fit using all predictors listed above to a logistic regression with only age entered as a linear term. We compute the AUC for the resulting SL/logistic regression fits on the 80% of the sample, both on the same data used to estimate SL/logistic regression models (training AUC), as well as a more realistic assessment by using the test set (the left out 20% of the available data used to calculate the testing set AUC).

To interpret the final prediction model generated by the SL fit, we use the permutation-based variable importance measure to identify variables that influence the SL model's prediction.[30] This is performed by permuting the predictor variables one at a time (keeping the other variables fixed) and measuring the magnitude of the decline on the predictive performance (as measured by the change in the average negative log-likelihood). This provides a list of variables ranked by the relative importance to prediction fit but does not provide information on the variable impact on mortality, which led us to another measure of RR using targeted maximum likelihood estimation (TMLE).

### Pre-existing condition RR estimate through TMLE

For pre-existing conditions, we estimated a different variable importance measure that is not focused on prediction accuracy but on estimating potential impacts of pre-existing conditions on mortality risk. The impact is estimated by the RR of adjusted means (adjusted for baseline confounders) for the population if everyone had the specific pre-existing condition of interest (the numerator) versus the same population where no one has the specific pre-existing condition (the denominator). To estimate RRs, we used cross-validated targeted minimum loss-based estimation (cross-validated TMLE). TMLE is a semiparametric substitution estimator that has shown to be asymptotically efficient (unlike the inverse probability of treatment-weighting estimators[34]). It also has some robustness advantages over other semiparametric-efficient approaches, such as augmented inverse probability weighting. TMLE estimates parameters that, under certain assumptions, can be interpreted as potential causal impacts of these factors on mortality, in our case, in the form of a causal RR. Our ensemble machine learning is optimised for prediction, but it does not directly provide measures of individual variable importance. We augmented our prior SL analysis using the TMLE to generate interpretable estimates of variable impact with robust SEs.[35–37] Both analyses using SL and TMLE are conducted in programming language R; the code used to conduct this analysis is publicly available on GitHub (link: https://github.com/ldliao/mexPred).

## RESULTS

Descriptive results show the age distribution of laboratory-confirmed patients across the three different epidemic phases (online supplemental figure S2). Phases 1 and 2 have similar distributions, and there are more young people (under 30) in phase 3. The six most prevalent pre-existing conditions are hypertension, obesity, diabetes, smoking, asthma and renal disease (online supplemental figure S3). The prevalence of all pre-existing conditions decreased over the three phases, and prevalence of hypertension, obesity and diabetes was drastically reduced in phase 3.

### SL prediction

SL fit has high prediction accuracy on the testing set (AUC: 0.907 (95% CI 0.905–0.908)) (table 2). The SL fit leverages multiple machine learning models: the XGBoost models, generalised additive model and random forest for prediction (online supplemental table S2). The simple logistic regression has a lower AUC (testing AUC: 0.874 (95% CI 0.872–0.876)) than the SL fit, as expected, as it only uses age as a predictor (table 2). However, the simple model is already highly predictive, and the difference is small yet significant. The logistic regression model overpredicts mortality risks for those roughly above age 75 compared with the SL prediction (figure 1). In line with the simple age-only logistic regression model, permuted variable importance on the SL fit shows, while holding other variables constant, age is consistently the most important for SL prediction in average mortality risk (online supplemental figure S4 and table S3). Having multiple comorbidities can dramatically increase risk for those individuals (figure 2).

**Table 2** Prediction results

| | All time (March 2020 to November 2021) AUC (95% CI) | Phase 1 (March 2020 to October 2020) AUC (95% CI) | Phase 2 (November 2020 to March 2021) AUC (95% CI) | Phase 3 (April 2021 to November 2021) AUC (95% CI) |
|---|---|---|---|---|
| Super learner fit | Training: 0.916 (0.915–0.917) Testing: 0.907 (0.905–0.908) | Training: 0.887 (0.885–0.888) Testing: 0.873 (0.870–0.876) | Training: 0.904 (0.903–0.906) Testing: 0.895 (0.892–0.897) | Training: 0.914 (0.913–0.916) Testing: 0.906 (0.902–0.909) |
| Age-only logistic regression fit | Training: 0.874 (0.873–0.875) Testing: 0.874 (0.872–0.876) | Training: 0.845 (0.843–0.846) Testing: 0.846 (0.842–0.850) | Training: 0.868 (0.866–0.870) Testing: 0.871 (0.868–0.874) | Training: 0.867 (0.865–0.869) Testing: 0.871 (0.866–0.875) |

Training AUC refers to prediction results using only 80% of the data.
Testing AUC refers to prediction results using only 20% left out of the data (disjoint from training). The testing set is not used for model development.
AUC, area under the receiver operating characteristic curve.

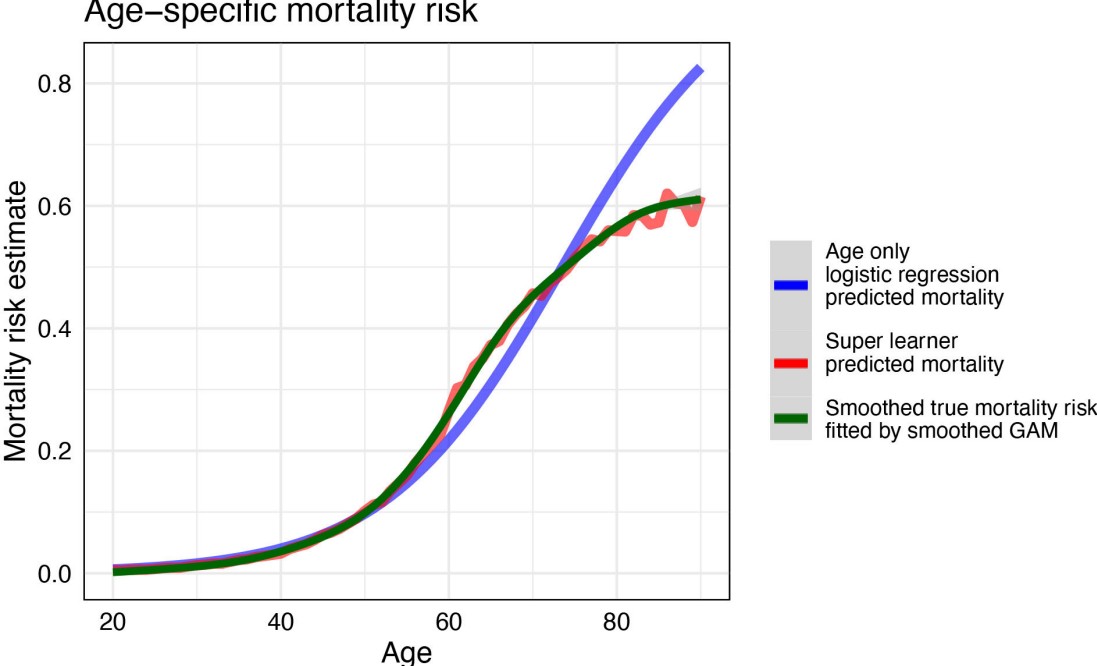

**Figure 1** Mortality risk prediction comparing age-only logistic regression and super learner. The smoothed true mortality risk curve is generated using a generalised additive model (GAM) with integrated smoothness estimation fitted with cubic splines.

### RRs of pre-existing conditions

To assess the impact of each pre-existing condition, we estimate their respective RRs of mortality, adjusting for demographic variables. We report the estimated RRs in table 3, ordered by impact (most to least) (online supplemental figure S5). The RRs compare the expected risk if all patients have the pre-existing condition (with) versus if all patients do not have the condition (without). The highest impact pre-existing condition is renal disease (RR: 3.783 (95% CI 3.705, 3.862)); diabetes, obesity and hypertension also have high impact individually (RR: 1.432–1.847). Minimal differences between the risk estimates are shown for smoking and asthma (RR: 1.049 and 1.037, respectively).

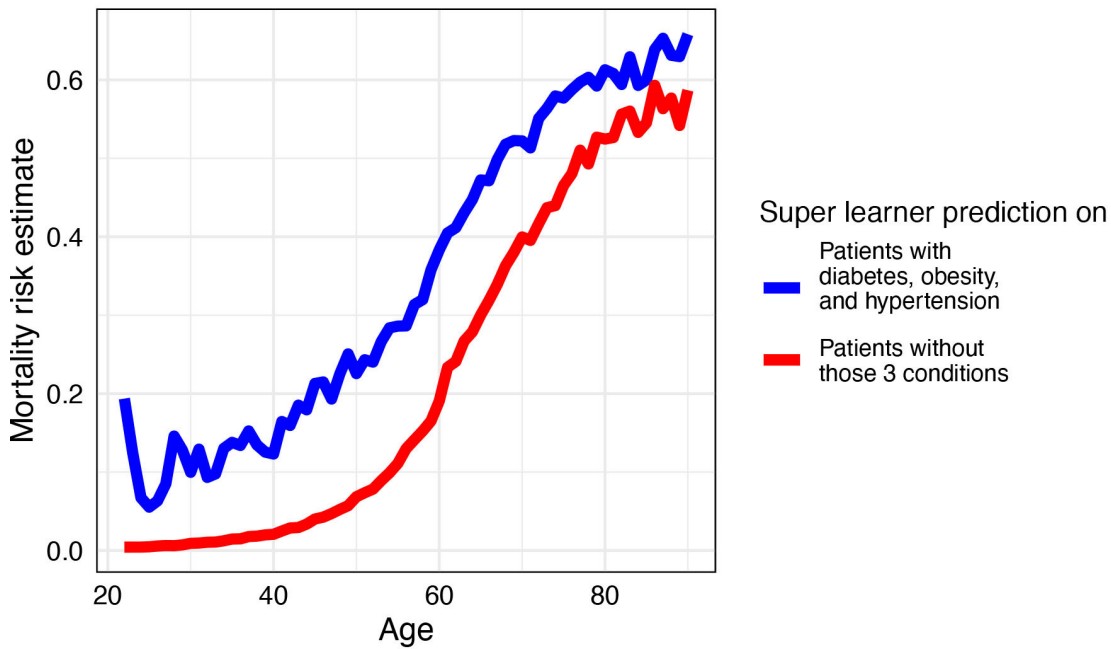

**Figure 2** Super learner predicted mortality risk averaged by specific age in two subgroups: those having all obesity, diabetes and hypertension pre-existing conditions versus those without.

**Table 3** Targeted maximum likelihood estimation relative risk results for each pre-existing condition

| | All time (March 2020 to November 2021) Relative risk (95% CI) | Phase 1 (March 2020 to October 2020) Relative risk (95% CI) | Phase 2 (November 2020 to March 2021) Relative risk (95% CI) | Phase 3 (April 2021 to November 2021) Relative risk (95% CI) |
|---|---|---|---|---|
| Renal disease | 3.783 (3.705, 3.862) | 2.588 (2.521, 2.657) | 2.994 (2.910, 3.080) | 6.638 (6.361, 6.927) |
| Diabetes | 1.847 (1.820, 1.875) | 1.536 (1.508, 1.566) | 1.594 (1.564, 1.625) | 2.508 (2.423, 2.596) |
| Hypertension | 1.745 (1.721, 1.770) | 1.427 (1.402, 1.452) | 1.500 (1.474, 1.527) | 2.356 (2.279, 2.436) |
| Obesity | 1.432 (1.417, 1.447) | 1.269 (1.249, 1.288) | 1.259 (1.239, 1.279) | 1.794 (1.750, 1.840) |
| Smoking | 1.049 (1.030, 1.068) | 1.001 (0.975, 1.028) | 0.992 (0.966, 1.018) | 1.158 (1.107, 1.210) |
| Asthma | 1.037 (1.002, 1.073) | 0.941 (0.895, 0.989) | 0.942 (0.892, 0.995) | 1.223 (1.134, 1.319) |

The phase analyses indicate pre-existing conditions are especially important in phase 3. Phases 1 and 2 are very similar in terms of both risk prediction and adjusted mortality risk estimates. However, in phase 3, age is less important in prediction (online supplemental table S3) and RRs drastically increase for every comorbidity. The adjusted risks show the decrease for each pre-existing condition in phase 3 (online supplemental table S4).

## DISCUSSION

Our analysis of (>1.4 million) laboratory-confirmed patients with COVID-19 demonstrates that age is by far the most important predictor of average mortality. For those patients with renal disease, diabetes, hypertension or obesity, having the comorbidity further increases their risk of mortality. A patient with diabetes, hypertension and obesity is roughly comparable to a patient 20 years older with none of the conditions, based on the predicted mortality (figure 2). Thus, having a comorbidity increases risk of mortality and should be considered at any age. The reason that comorbidities add little to the predictive power at younger ages is that hypertension and diabetes are age related and the reported onset is often for those over 30, so the pre-existing conditions are far less prevalent.

Our prediction results using machine learning methods predict better than previous studies, and we demonstrated the feasibility and robustness of using machine learning methods targeted for prediction and variable impact. SL model prediction has an AUC of 0.907, which is higher than any previous Mexican study (AUCs from 0.634 to 0.824).[9 38] Although age has been well reported by previous studies as important,[6 38 39] our analysis is more robust because we do not assume a prespecified functional relationship between the explanatory variables and the predicted variable, and thereby avoid any arbitrary groupings into age categories. Moreover, since those above age 60 have a higher prevalence of comorbidities, relying on simple logistic regression models can greatly overpredict the average mortality risk for the elder patients. Our study applies TMLE to estimate the adjusted mortality risk ratios for each comorbidity to provide more robust impact estimates that respect time ordering and account for background variables.

We find consistent results of comorbidities compared with previous studies, and present phase analyses highlighting the changes in RRs over time. Previous results from logistic regressions indicated ORs of 1.458–2.48 for renal disease, 1.237–1.74 for diabetes, 1.173–1.47 for obesity, 1.194–1.315 for hypertension, 0.852–1.02 for smoking and 0.74–1.420 for asthma.[38–40] Although our analysis is generally consistent with previous findings, our RR estimations have less uncertainty. Renal disease has the greatest impact on mortality, followed by diabetes, hypertension and obesity; smoking and asthma have negligible impact on mortality risk.

This phase-specific analysis produced a seemingly paradoxical finding. The impact of comorbidities on predicted mortality decreased with time (primarily between the second and third waves), but the RR on mortality dramatically increased for the same conditions (online supplemental table S4 and figure S5). The apparent explanation is that mortality risk for people without the comorbidities fell faster than for people with them, increasing the RR. The decrease in mortality risk is multifactorial and includes a decrease in susceptibility over time (due to prior infection and vaccination), improved treatment, enhanced healthcare response and opportunity to be admitted to a hospital or intensive care unit, and less virulent viral subtypes. This implies that as herd immunity increases, medical resources should focus even more on protecting vulnerable people at older age and those with comorbidities since they are even more likely to experience severe outcomes compared with those who are younger and/or healthier.

Readers should be cautious about extrapolating our findings to other populations. Although our sample is large and includes patients from all parts of Mexico, most of the patients were IMSS beneficiaries. In order to access IMSS health services, patients require to: (a) be a formal sector worker or retired, (b) be a direct dependent of such an employee, (c) be a bachelor or postgraduate student in a public institution and (d) voluntarily enrol by paying a fee. Thus, the IMSS population skews towards the upper half of the income distribution.

Populations without similar access to health services may have different results. It is also important to consider the potential impact of data quality. Pre-existing conditions were self-reported and likely also inconsistently recorded, perhaps in systematic ways that could have biased the results. For example, if people with severe diabetes were more likely to report diabetes as a pre-existing condition, we may overestimate the impact of diabetes on mortality.

It is also important to consider what predictive variables are included in this model. We sought to predict risk for an individual in the population using their characteristics prior to infection. In other words, what is this person's risk of death from COVID-19 if they were to be infected? The answer to this question best informs the question of who should be prioritised for protection against infection or for early therapeutic interventions following infection. It does not attempt to predict the likely mortality of a patient who presents to the health services with COVID-19 because information about that patient's severity of their COVID-19-related symptoms will represent important additional predictors of their mortality risk.

**Author affiliations**
[1]Division of Biostatistics, University of California Berkeley, Berkeley, California, USA
[2]Center for Policy, Population and Health Research, School of Medicine, Universidad Nacional Autónoma de México, Ciudad de México, Mexico
[3]Micron Technology, Boise, Idaho, USA
[4]College of Computing, Data Science, and Society, University of California Berkeley, Berkeley, California, USA
[5]Department of Electrical Engineering and Computer Sciences, University of California, Berkeley, Berkeley, California, USA
[6]Instituto Mexicano del Seguro Social, Ciudad de México, Mexico
[7]Division of Health Policy and Management, University of California, Berkeley, Berkeley, California, USA
[8]School of Public Health, University of Washington, Seattle, Washington, USA
[9]Instituto Nacional de Salud Pública, Cuernavaca, MOR, Mexico

**Acknowledgements** We thank the staff of C3.ai DTI for their technical support and our colleagues at the University of California, Berkeley, the Mexican National Autonomous University and the Mexican Social Security Institute (IMSS) for all of the administrative and technical support that has allowed this collaboration to flourish.

**Contributors** LDL and AEH contributed to the study design and methodology. AJ, YY and IdJA-M contributed to data acquisition. LDL, YY and KK contributed to data cleaning. LDL led the data analysis and visualisation. LDL, AEH, JPG and SMB interpreted the results. LDL drafted the manuscript with support from RG on literature search. AEH and SMB significantly contributed to the revision of the manuscript. All authors participated in review and edited the manuscript; all authors have read and approved the final manuscript. All authors had full access to all the data in the study and accepted responsibility to submit for publication. All authors take responsibility for the integrity of the data and the accuracy of the data analysis. SMB serves as the guarantor, who accepts full responsibility for the finished work and the conduct of the study, had access to the data, and controlled the decision to publish.

**Funding** C3.ai Digital Transformation Institute, National Science Foundation (award/grant number: DGE 2146752), Bill & Melinda Gates Foundation (award/grant number: OPP1165144).

**Competing interests** None declared.

**Patient and public involvement** Patients and/or the public were not involved in the design, or conduct, or reporting, or dissemination plans of this research.

**Patient consent for publication** Not applicable.

**Ethics approval** This data-only study was approved on 4 November 2020 by the Scientific Research National Committee (Mexican Social Security Institute, R-2020-785-165). The University of California Berkeley Institutional Review Board (IRB) determined that the project was exempt from IRB approval.

**Provenance and peer review** Not commissioned; externally peer reviewed.

**Data availability statement** No data are available. The study was conducted using confidential patient records subject to strict access controls and we are therefore unable to share the data that were used for this study.

**ORCID iDs**
Lauren D Liao http://orcid.org/0000-0003-4697-6909
Juan Pablo Gutierrez http://orcid.org/0000-0002-0557-5562
Stefano M Bertozzi http://orcid.org/0000-0002-1723-7085

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
