## [Reviewer comments · BMJ Open]

ARTICLE DETAILS

TITLE (PROVISIONAL)	Who is most at risk of dying if infected with SARS-CoV-2? A mortality risk factor analysis using machine learning of COVID-19 patients over time: a large population-based cohort study in Mexico.
AUTHORS	Liao, Lauren; Hubbard, Alan; Gutierrez, Juan Pablo; Juárez, Arturo; Kikkawa, Kendall; Gupta, Ronit; Yarmolich, Yana; de Jesús Ascencio-Montiel, Iván; Bertozzi, Stefano

VERSION 1 – REVIEW

REVIEWER	Di Castelnuovo, Augusto Fondazione GP2
REVIEW RETURNED	09-Mar-2023

GENERAL COMMENTS	I appreciated the manuscript; however, I have some concerns: 1) The authors omitted information on timing of data collection.: how many years passed from data collection (about pre-existing conditions) to positivity to Sars-Cov-2? Please, add these pieces of information2) The authors used mortality for any cause as outcome. They omitted information on timing of death. How many years passed from positivity to Sars-Cov-2 to death or staying alive? Please, add these pieces of information3) Research ethics (e.g. participant consent, ethics approval) are not addressed4) Line 57: ">60 M" is misleading. The authors started from a sample large 4.5M at most5) Lines 66-67 need of a bibliographic reference6) Lines 93-101 list similar data from several countries. The list is inevitably non-exhaustive. Literature is plenty of review and meta-analysis on the topic. I suggest to cite the most updated and complete of them7) Line 178. Why the authors used as benchmark a logistic model including only age? To my opinion, there is no reason to not include other potential predictors (pre-existing conditions etc.) in the logistic model. I know that "basic" logistic regression models, are not able to deal with non-linearity of effects or with interactions. But this is not a reasonable reason to use only age. Moreover, more complex logistic regression models are available, which are able to deal in some way with non-linearity (splines), and additive or multiplicative terms of interaction can be included in the model. Comparing a SL plenty of predictors with a logistic model with only age is non-sense. Superiority of SL should be demonstrated when comparing it against complex and complete logistic model8) What the authors does mean in line 205? What "follow-up"?9) Please, the authors should give information on software used for SL and TMLE10) Line 255: may the authors refer to previous studies not limited to
---

	Mexico?
REVIEWER	Hajra, Adrija Jacobi Medical Center, Internal Medicine
REVIEW RETURNED	04-Jun-2023
GENERAL COMMENTS	Thank you for the article. I would suggest some minor edits: The probability that someone infected with SARS-CoV-2 dies has varied enormously over time, among countries, and among population groups within countries-rewrite: The probability of mortality associated with SARS-CoV-2 infection had.... Page 5, line 33-Late stage CKD-any specific stage that can be mentioned? (like stage IIIb, IV) Though the authors have explained the methods and result in the main manuscript, on table-2 it would be clear to know the explanation of "training" and "testing".

VERSION 1 – AUTHOR RESPONSE

Reviewer: 1
Dr. Augusto Di Castelnuovo, Fondazione GP2

Comments to the Author:
I appreciated the manuscript; however, I have some concerns:

[Response]: We thank Dr. Di Castelnuovo for their detailed and thoughtful comments. Please find our point-by-point response to each comment below.

1. The authors omitted information on timing of data collection.: how many years passed from data collection (about pre-existing conditions) to positivity to Sars-Cov-2? Please, add these pieces of information

[Response]: We thank Dr. Di Castelnuovo for the helpful suggestion.

The questions about pre-existing conditions were asked of patients when they were suspected of being possible COVID cases and referred for testing for SARS-CoV-2. In other words, the data on pre-existing conditions and positivity are concurrent. We were not able to access the patients' preexisting outpatient medical records for independent verification of the pre-existing conditions.

This is reflected in the existing text on page 8 lines 178-180 manuscript with tracked changes. "Patients were asked at presentation about their pre-existing health conditions; these were not ascertained with reference to the patient's medical record, even for those patients insured by the IMSS."

2. The authors used mortality for any cause as outcome. They omitted information on timing of death. How many years passed from positivity to Sars-Cov-2 to death or staying alive? Please, add these pieces of information

[Response]: We thank Dr. Di Castelnuovo for the comment. We did not intend to imply that we used all-cause mortality as our outcome measure. The mortality data comes from the SINOLAVE database and the mortality reported in SINOLAVE is COVID-related mortality as determined by the health providers registering the cause of death. This suggests that only COVID deaths that are proximally related to acute COVID illness are likely to be captured in the database. While this would exclude deaths from long-term complications of COVID, the overwhelming majority of COVID deaths occur during the acute illness episode. We have tried to clarify the text so that other readers don't interpret

the mortality as all-cause mortality.

This clarification is reflected in the manuscript with tracked changes on page 8 line 183-185.

“Mexican state in which the patient received care, COVID-19 test results (from both polymerase chain reaction (PCR) tests and antigen tests), the month that the patient initiated care, and COVID-related mortality. The outcome, death, is ascertained as COVID-related mortality within this study period between March 2020 and November 2021; we only consider deaths after patients initiated care.”

3. Research ethics (e.g. participant consent, ethics approval) are not addressed

[Response]: We thank Dr. Di Castelnuovo for the comment. The research ethics and the approval are written on page 20 line 420 to 424 in the manuscript with tracked changes. If additional ethics approval is needed for this study as per the BMJ Open editorial policy, we are happy to provide further details. The Berkeley IRB determined that the project was exempt from IRB approval because it worked with deidentified, previously collected data that was not reported at the level of individual patients. The existing text reads as follows:

Ethics approval

This data-only study was approved on November 4th, 2020, by the Scientific Research National Committee (Mexican Social Security Institute) with R-2020-785-165. The University of California, Berkeley Institutional Review Board (IRB) determined that the project was exempt from IRB approval.

4. Line 57: “>60 M” is misleading. The authors started from a sample large 4.5M at most

[Response]: We thank Dr. Di Castelnuovo for his comment and agree with him. We have changed the text to refer to the number of analytical sample and not the overall sample frame.

We revised this in the manuscript with tracked changes for the first bullet point, on page 4 line 58.

“Compared to previous studies showing age and the presence of pre-existing conditions as important predictors of mortality, our study leveraged more powerful statistical approaches by examining mortality risk in a very large population (> 4.5 M).”

5. Lines 66-67 need of a bibliographic reference

[Response]: We thank Dr. Di Castelnuovo for the comment. We have updated following text with citation.

We revised the text in the manuscript with tracked changes on page 5 line 68-69.

“The probability of mortality associated with SARS-CoV-2 infection has varied enormously over time, among countries, and among population groups within countries [1].”

The citation refers to the following article.

1. Shang W, Wang Y, Yuan J, et al. Global Excess Mortality during COVID-19 Pandemic: A Systematic Review and Meta-Analysis. *Vaccines (Basel)* 2022;10. doi:10.3390/vaccines10101702

6. Lines 93-101 list similar data from several countries. The list is inevitably non-exhaustive. Literature is plenty of review and meta-analysis on the topic. I suggest to cite the most updated and complete of them

[Response]: We thank Dr. Di Castelnuovo for the comment. Given how dynamic the existing available relevant published literature, we have updated the references to reflect more recent work. Therefore, we updated the text with multiple citations to reflect recent reviews and discussions on this topic

We revised the text in the manuscript with tracked changes on page 6 line 118-121.

“... A Canadian study reported dementia, chronic kidney disease, cardiovascular disease, diabetes, COPD, severe mental illness, organ transplant, hypertension, and cancer to be significant predictors of mortality [15]. These studies represent a non-exhaustive list of research studying COVID-19 risk factors and mortality. Recent meta-analyses and systematic reviews find significant mortality attributed to these pre-existing conditions [1, 16-18]. Our goal in this study is not only to predict

mortality using demographic factors and comorbidities, but to show how those predictions change over time in this rapidly evolving pandemic.”

The citations associated with this comment refer to the following articles.

1. Shang W, Wang Y, Yuan J, et al. Global Excess Mortality during COVID-19 Pandemic: A Systematic Review and Meta-Analysis. *Vaccines (Basel)* 2022;10. doi:10.3390/vaccines10101702
16. Sawadogo W, Tsegaye M, Gizaw A, et al. Overweight and obesity as risk factors for COVID-19-associated hospitalisations and death: systematic review and meta-analysis. *BMJ Nutr Prev Health* 2022;5:10–8.
17. Kastora S, Patel M, Carter B, et al. Impact of diabetes on COVID-19 mortality and hospital outcomes from a global perspective: An umbrella systematic review and meta-analysis. *Endocrinol Diabetes Metab* 2022;5:e00338.
18. Akbari A, Fathabadi A, Razmi M, et al. Characteristics, risk factors, and outcomes associated with readmission in COVID-19 patients: A systematic review and meta-analysis. *Am J Emerg Med* 2022;52:166–73.

7. Line 178. Why the authors used as benchmark a logistic model including only age? To my opinion, there is no reason to not include other potential predictors (pre-existing conditions etc.) in the logistic model. I know that “basic” logistic regression models, are not able to deal with non-linearity of effects or with interactions. But this is not a reasonable reason to use only age. Moreover, more complex logistic regression models are available, which are able to deal in some way with non-linearity (splines), and additive or multiplicative terms of interaction can be included in the model. Comparing a SL plenty of predictors with a logistic model with only age is non-sense. Superiority of SL should be demonstrated when comparing it against complex and complete logistic model

[Response]: We thank Dr. Di Castelnuovo for the valuable comment and regret any confusion. Indeed, more sophisticated logistic regression can perform comparably to a complex super learner (SL). However, our goal is not to demonstrate the superiority of the SL to a basic logistic regression that only includes an age term. Our purpose of including this simple logistic regression is to show that age alone is already highly predictive, especially at younger ages. We also show how additional predictability can be obtained as seen in the SL fit, specifically for those above age 75 (Fig. 1).

From our analysis, using only the simple age-only logistic regression performs well with testing AUC 0.846-0.871, as shown in Table 2. The difference in AUC between the logistic regression and SL fit is small, yet important. We use the SL to fit the data more flexibly by leveraging multiple machine learning algorithms and enable selection of algorithms that is potentially more sophisticated than logistic regression. Since we included the fully-specified logistic regression as a candidate in the SL library, we can examine the mean squared errors for a comparison against other models (shown in Supplemental Table S2). From the Supplemental Table S2, we can see that the fully-specified version of the logistic regression has a mean squared error marginally higher (slightly lower performance) than the algorithms chosen by the SL.

We revised the text in the manuscript with tracked changes on pages 12-13 line 282-289. “SL fit has high prediction accuracy on the testing set (AUC: 0.907 (95% CI: (0.905-0.908))) (Table 2). The SL fit leverages multiple machine learning models: the XGBoost models, generalized additive model, and random forest for prediction (Supplemental Table S2). The simple logistic regression has a lower AUC (testing AUC: 0.874 (95% CI: (0.872-0.876))) than the SL fit, as expected, as it only uses age as a predictor (Table 2). However, the simple model is already highly predictive, and the difference is small yet significant. The logistic regression model overpredicts mortality risks for those roughly above age 75 compared to the SL prediction (Fig. 1). In line with the simple age-only logistic regression model, permuted variable importance on the SL fit shows, while holding other variables constant, age is consistently the most important for SL prediction in average mortality risk (Supplemental Figure S4 and Table S3). Having multiple comorbidities can dramatically increase risk for those individuals (Fig. 2).”

8. What the authors does mean in line 205? What “follow-up”?

[Response]: We thank Dr. Di Castelnuovo for the comment and regret the ambiguous language. We have revised the language to clarify that we augmented our previous SL prediction step with TMLE for bias reduction and robust inference.

We revised the text in the manuscript with tracked changes on page 12 line 268.

“We augmented our prior SL analysis using the TMLE to generate interpretable estimates of variable impact with robust standard errors [35-37].”

9. Please, the authors should give information on software used for SL and TMLE

[Response]: We thank Dr. Di Castelnuovo for the helpful suggestion and agree that the impact of our manuscript would be greater if our annotated code were available. In name of transparency and reproducibility, we have created a public repository with the R code that was used to generate the analysis on GitHub. Others who wish to conduct similar analysis can consult the public repository.

We revised the text in the manuscript with tracked changes on page 12 line 269-271.

“Both analyses using SL and TMLE are conducted in programming language R; the code used to conduct this analysis is publicly available on GitHub (link: <https://github.com/ldliao/mexPred>).”

10. Line 255: may the authors refer to previous studies not limited to Mexico?

[Response]: We thank Dr. Di Castelnuovo for this comment. Prediction AUCs from prior studies outside of Mexico may not be comparable due to context specific surveillance methods and populations differences. Comparing different prediction AUCs for other locations is beyond the scope of our study. If Dr. Di Castelnuovo is more concerned with the existing evidence supporting the optimality of the SL for prediction and targeted maximum likelihood estimation for causal parameters and variable importance, it has been well-demonstrated in dozens of publications that include both theoretical support as well as empirical side-by-side comparisons of these methods to other competing approaches. Though we have limited the technical discussion of the justification of these methodologies, we could add more text and citations justifying their use if desired.

Reviewer: 2

Dr. Adrija Hajra, Jacobi Medical Center

Comments to the Author:

Thank you for the article. I would suggest some minor edits:

[Response]: We thank Dr. Hajra for their valuable comments. Please find our point-by-point response to each comment below.

1. The probability that someone infected with SARS-CoV-2 dies has varied enormously over time, among countries, and among population groups within countries-rewrite: The probability of mortality associated with SARS-CoV-2 infection had....

[Response]: We thank Dr. Hajra for this suggestion. We have incorporated this writing into our revision.

We revised the text in the manuscript with tracked changes on page 5 line 68-69

“The probability of mortality associated with SARS-CoV-2 infection has varied enormously over time, among countries, and among population groups within countries [1].”

2. Page 5, line 33-Late stage CKD-any specific stage that can be mentioned? (like stage IIIb, IV)

[Response]: We thank Dr. Hajra for this comment. Unfortunately, the SINOLAVE database does not include any information on the stage of CKD, and unfortunately, we were not able to access the patients' ambulatory electronic medical records data.

3. Though the authors have explained the methods and result in the main manuscript, on table-2 it would be clear to know the explanation of "training" and "testing".

[Response]: We thank Dr. Hajra for this helpful comment. Further revisions for more detailed explanation of “training” and “testing” sets in Table 2 are added.

We revised the caption of Table 2 in the manuscript with tracked changes on page 13 line 295.
“AUC: area under the receiver operating characteristic curve; CI: confidence interval
Training AUC refers to prediction results using only 80% of the data.
Testing AUC refers to prediction results using only 20% left-out of the data (disjoint from training). The testing set is not used for model development.”

VERSION 2 – REVIEW

REVIEWER	Di Castelnuovo, Augusto Fondazione GP2
REVIEW RETURNED	30-Aug-2023
GENERAL COMMENTS	The authors provided a satisfactory revision